# Different Types of Meatballs Enriched with Wild Thyme/Lemon Balm Aqueous Extract—Complex Characterization

**DOI:** 10.3390/molecules27123920

**Published:** 2022-06-18

**Authors:** Luiza-Andreea Tănase (Butnariu), Oana-Viorela Nistor, Doina-Georgeta Andronoiu, Gabriel-Dănuț Mocanu, Andreea Veronica Botezatu Dediu, Elisabeta Botez

**Affiliations:** 1Faculty of Food Science and Engineering, “Dunărea de Jos” University of Galați, 111 Domnească Street, 800201 Galați, Romania; luiza.tanase@ugal.ro (L.-A.T.); georgeta.andronoiu@ugal.ro (D.-G.A.); danut.mocanu@ugal.ro (G.-D.M.); elisabeta.botez@ugal.ro (E.B.); 2Department of Chemistry, Physics and Environment, Faculty of Sciences and Environment, “Dunărea de Jos” University of Galați, 111 Domnească Street, 800201 Galați, Romania; andreea.botezatu@ugal.ro

**Keywords:** meatballs, aqueous extract, galactogogue, color parameters, FT-IR analysis

## Abstract

In the context of the increasing lactation problems among breastfeeding women, the development of a healthy lifestyle is needed. Different variants of pork, turkey, and beef meatballs, with added lemon balm (*Melissa officinalis* L.) and wild thyme (*Thymus serpyllum* L.) aqueous extract (6%), were obtained. These herbs were selected and used due to their antioxidant, antimicrobial, and lactogenic potential. Two thermal treatments, hot air convection (180 °C) and steam convection (94 °C), were applied for meatballs processing. The obtained meatballs were further subjected to a complex characterization. The functionality of the plant extracts was proved by the values of total content of polyphenols (2.69 ± 0.02 mg AG/g dw) and flavonoids (3.03 ± 0.24 mg EQ/g dw). FT-IR analysis confirmed the presence of trans-anethole and estragole at 1507–1508 cm^−1^ and 1635–1638 cm^−1^, respectively. Costumers’ overall acceptance had a score above 5.5 for all samples, on a scale of 1 to 9. Further analysis and human trials should be considered regarding the use of lactogenic herbs, given their health benefits and availability.

## 1. Introduction

Nowadays, consumers’ hectic schedules have contributed to the development of a new trend towards the consumption of ready-to-eat (RTE) products. Even so, to be able to be chosen and consumed, a product should accomplish various requirements after quality and safety, such as diversity and pro-health benefits. There are numerous herbs (thyme, lemon balm, star anise, fennel, fenugreek, and many others) known to improve different health conditions (depression, cardiovascular diseases and diabetes type 1 or 2; reduction of obesity, cholesterol, and triglycerides levels), mainly due to their polyphenols content. The plant phenolic content may act as reducers (free radical terminators), hydrogen donors, singlet oxygen quenchers, and metal chelators due to their antioxidant properties [1]. In addition, the usage of synthetic antioxidants is one of the major approaches for preventing oxidative reactions, and for extending the shelf life of meat products. However, they have numerous unhealthy connotations. Therefore, modern scientific research shifted towards various natural alternatives, focusing on plant extracts [2]. *Melissa officinalis* L. is an edible and medicinal plant belonging to the mint family *Lamiaceae*. It is native to Central Europe, the Mediterranean region, and Central Asia, having a well-documented ethnomedical reputation, especially as a sedative, antipyretic, antispasmodic, antihypertensive, anti-Alzheimer’s, and antiseptic [3]. Above all, [4] stated that the use of lemon balm could be an effective and harmless measure for the treatment of infantile colic, both in breastfeeding and even for bottle-fed infants. The genus *Thymus* L. is one of the largest and most economically important genera in the *Lamiaceae* family [5]. *Thymus serpyllum* is well-known as wild thyme and it is native to Mediterranean Europe and North Africa [6]. Wild thyme is traditionally used as an expectorant and antimicrobial in upper respiratory tract infections due to its bronchoantispasmodic, expectorant, and antibacterial activity [7]. Furthermore, thyme is one of the most commonly used plants in the postpartum period [8].

Aqueous extracts are the most popular forms of herbal processing, whether it is infusion, decoction, or even maceration. The herbal aqueous extracts are combined with three different types of meat (pork, turkey, and beef) to obtain an RTE product designed to enhance lactation among breastfeeding women. Meat is rich in proteins with a high biological value, which makes it essential for a well-balanced diet [9]. Traditionally, pork meatballs contained an average of 20–30% of fat. Saturated fat was associated with increased risk for several conditions including obesity, type 2 diabetes, and cardiovascular diseases [10]. As a consequence, tenderloin was selected, which is known to have a high protein intake and a very low-fat content [11]. Traditional heat treatments (boiling or frying) result in the formation of undesirable compounds and a significant increase in oxidative reactions in meat lipids [12], so less harmful processing methods such as hot air and steam convection were chosen.

The objective of this research was to develop and characterize meatballs enriched with aqueous extracts of lemon balm or wild thyme. Twelve meatball variants were evaluated for bioactive content, FT-IR determination, color measurements, and sensorial analysis.

## 2. Results and Discussion

### 2.1. Antioxidant Activity and Bioactive Compounds

Many of the therapeutic actions of numerous plants are attributed to their biologically active components, such as flavonoids and phenolic acids, which possess antioxidant activity [13]. In addition, natural antioxidants are multifunctional and are gaining popularity as an alternative to synthetic antioxidants in the prevention of oxidation in complex food systems [14].

In Table 1, the main phytochemicals from meatballs with herbal aqueous extract addition are presented. 

From Table 1, it can be observed that the highest value is exhibited by ERPC, which is represented by pork meatball enriched with lemon balm aqueous extract, obtained by hot air convection. In the study conducted by [14], regarding the batters enriched with *Hypericum perforatum* L., lower values for antioxidant activity were reported, between 0.171 ± 0.002 µM Trolox/g dw and 0.440 ± 0.021 µM Trolox/g dw. In terms of the total phenolic content, the values varied between 1.38 ± 0.04 and 2.69 ± 0.02 mg GAE/g dw. In addition, in terms of the total phenolic content, the values registered were 6 times higher than those obtained by [15] for some minced beef and chicken products with the addition of pomegranate seed extract, obtained by four cooking methods (pan cooking, oven roasting, charcoal barbecue, and deep-fat frying). Furthermore, as expected, the total flavonoid content ranged in the same trend related to the antioxidant activity values.

### 2.2. FT-IR Analysis

Trans-anethole and estragole are considered two active estrogenic agents according to [16], mostly found in galactogogue plants. The FT-IR analysis of aqueous extract enriched meatballs was aimed at revealing the presence of specific compounds or functional groups of estrogenic agents in minced meat products. The FT-IR spectra of all samples were recorded in the spectral range of 400–4000 cm^−1^, which are represented in Figure 1.

The most prominent peak (3274 cm^−1^) of the FT-IR spectra for the analyzed samples is attributed to water (3200–3550 cm^−1^), which can be due the composition of the raw material containing between 72–75% water [17,18,19]. The appearance of other important peaks, 1558 cm^−1^, 1540 cm^−1^ and 1507 cm^−1^ was found, according to [19], in the protein region amide II (1500–1600 cm^−1^). The slope between 2800 cm^−1^ and 3500 cm^−1^ is attributed to lipids (cholesterol, phospholipids) and creatine [19]. Notable absorption bands found in the region between 1700–1500 cm^−1^ are dominated by the stretching vibrations of the C=O and C-N group belonging to the amide groups, which include the peptide bonds in proteins [20]. The amide I band is the most prevalent band in the IR spectrum, resulting from the stretching vibration of the C=O group with the involvement of N-H groups from protein folding. Even so, the amide I band is particularly sensitive to changes in secondary protein structure, being identified in the range of 1623–1637 cm^−1^, as mentioned by [21]. Similar changes were attributed to minced meat samples through the identification of two peaks, 1635 cm^−1^ and 1624 cm^−1^.

According to [21], in the range of 1350–1200 cm^−1^ can be identified amide III resulting from the folding in the plane of the N-H bond and the C-N tensile vibration. Similar to previous findings, amide III can be found in minced meat products by the presence of bands at 1339 cm^−1^ and 1239 cm^−1^. In addition, according to [22], the stretching vibrations of C=O evidenced in the range of 1650–1600 cm^−1^ are associated with ketone groups, while the stretching vibrations C=C in the range of 1510–1450 cm^−1^ belong to aromatic compounds. Absorption bands characteristic of ketone groups at 1635 cm^1^ and 1638 cm^−1^ were identified for both minced meat samples and estragole. The peak corresponding to the value of 1507–1508 cm^−1^ was found both in the case of meat samples and trans-anethole. This is due to the presence of aromatic compounds in both matrices analyzed.

In accordance with [22], in the region 2935–2915 cm^−1^ asymmetric extension -CH (CH2) was presented related to the saturated aliphatic compounds. These vibrations were also found in minced meat samples and in the case of trans-anethole at 2922 cm^−1^ and 2925 cm^−1^, respectively.

In addition, [23,24] report the asymmetric tensile vibration of the methylene group (–CH2) at 2924 cm^−1^. The work of [25] reported similar findings for tensile vibrations of the C-O-C connection, in the interval 1200–1250 cm^1^. The band 1241–1242 cm^−1^ was determined both in the minced meat samples and in the standard trans-anethole and estragole samples, which could explain the presence of these galactogogue compounds in meatballs. In addition, the specific trans-anethole peak identified at 1440 cm^−1^ was found with a variation at 1436 cm^−1^ in meatballs. The range of 1450–1400 cm^−1^ is specific to the presence of stretching vibrations of the carbonyl group (C=O bond), as referred to by [25].

Consequently, the constituent components, known to aid lactation, trans-anethole, and estragole, are found in meatballs with herbal aqueous extract addition.

### 2.3. In Vitro Release of Phenolic Compounds from Different Types of Meatballs

The meatballs were digested in simulated gastric (SGJ) and intestinal juices (SIJ), and the phenolic content was measured. The current work will help advance our understanding of the in vitro bioaccessibility of phenolic compounds released from various meat matrices.

Figure 2 shows the percentage of released phenolic compounds after two hours of simulated gastrointestinal digestion.

ERCC showed the highest stability in the simulated gastric phase, with a maximum phenol release of 1.95 ± 0.08% after 120 min of digestion, out of all the samples studied (Figure 2a). ECPA released a substantial content of phenols from the matrix in the case of minced pork meatballs, reaching a maximum of 9.33 ± 0.10% after 120 min of digestion. Phenols in SGJ reveal a slightly increase (5.89 ± 0.01% and 1.95 ± 0.08%, respectively) in ERCA and ERCC meatballs when it comes to turkey samples.

The phenolic content from meatballs was released into the intestinal environment (Figure 2b). The negative downward slope generated by the variation of total polyphenols content in the SGJ can be traced to the samples obtained from minced beef compared to other types of meat. Even so, a significant release of phenols from the matrix was observed in the intestinal phase for ECVA, with a maximum of 29.89 ± 0.26% after 2 h of digestion.

### 2.4. Color Measurements

Meat color is a very important attribute due to the fact that it is the only one that could influence the consumers’ product acceptance [26]. Minced meat products with aqueous lemon balm/wild thyme extract were tested to assess the changes induced by the heat treatment (hot air and steam convection). Table 2 shows the effects of thermal treatments on the color parameters of meatballs enriched with lemon balm/wild thyme aqueous extract.

The color parameters’ values of the meatballs revealed changes in the *L**, *a**, and *b** parameters mainly based on the specific characteristics of the raw material. According to [27], the influence of herbal aqueous extracts among the color of meat products, presented in the scientific literature, is ambiguous. It could depend on the type of raw material used to obtain the extract, the extraction method, and the quantity of extract used. The degree of light dispersion generated by the muscles, which is determined by both the three-dimensional structure of the muscle network and the degree of refraction of the surrounding fluid, can affect the color of the meatball samples. The last assertion could be explained by the contribution of the globular protein termed myoglobin [28].

The heat treatment applied to minced meat products resulted in significant color changes (*p* < 0.05) in all types of raw materials tested: pork, turkey, and beef tenderloin. After both heat treatments, hot air and steam convection, the mean *L** values increased. In the case of minced pork products, mean *L** value raised with 50.6% by hot air convection and with 44.7% by steam convection, reported to the control sample. Brightness scored the highest values for baked turkey meatballs, 57.58 ± 0.33 and 57.75 ± 0.29, while steamed beef meatballs registered the lowest mean *L** values, 41.69 ± 1.07 and 41.37 ± 0.60. As a result, most minced beef products treated by hot air convection had higher mean *L** values than those processed using steam convection. The treatment’s temperature and thermal agent are important factors that generates changes of the meat color. Generally, high-temperature treatments (over 40 °C) and steam can have a significant impact on the color of the final product [29]. The data obtained are in agreement with [30], which states that a higher *L** value indicates a lighter color, which is desirable in order to assure customer approval. Closely related the sensory examination of meatballs sustains this assertion.

According to [28], the color parameters (*a**, *b**) tend to be strongly associated with the meat myoglobin’s pigment, while the brightness (*L**) is related to the structural characteristics of the muscle and the two combined determine the reflected seen light. However, the brightness values can also be explained by the fact that the diameter of the meat fiber shrinks after the heat treatment, causing more light to dissipate. Consistent with [31], the muscle fiber contraction creates large gaps between fibers and myofibrils appear to shrink, resulting in a denser protein structure. In addition, [26] reported that changes in meat opacity may be caused by globulin denaturation, heme group displacement, and aggregation of both myofibrillar and sarcoplasmic proteins. During processing, the beef color shifts from dark red to a light gray-pink with a light brown finish, changes associated with myoglobin denaturation according to [30].

Aside from turkey meatballs, the *a** values for beef and turkey meatballs obtained by hot air convection were significantly higher (*p* < 0.05) than those obtained by water vapor convection. According to [30], using water vapor convection caused more myoglobin degradation.

The effects of heat treatments on the color parameters of meatballs enriched with aqueous extract and storage at refrigeration temperature (4 °C) for 7 days are shown in Table 3.

Due to the presence of fat with high content of unsaturated fatty acids, meat products are susceptible to oxidation. Such processes occur during production and in later storage [27]. Therefore, the color parameters of the meatball samples were colorimetrically reevaluated after 7 days of refrigeration at 4 °C (Table 3). For both heat treatments, the values of the parameter *a** almost halved, meanwhile for the parameter *b** increased until 19.1% compared with the fresh samples. Thus, due to these changes of values, a significant rise (*p* < 0.05) of the yellow index was determined for all samples. The earlier research of [32] showed similar results regarding *YI* values, which have been correlated to the oxidation process happening during storage. Furthermore, according to [32], higher values for parameter *L** alongside extended storage time could be correlated with enhanced metmyoglobin formation. Thus, in the case of the refrigerated meatballs, there was a decrease in the amount of metmyoglobin, which is sustained by the decline of *L** values.

### 2.5. Sensory Evaluation

The sensory analysis assisted in determining whether or not the customers’ needs were met. The meatballs were rated on a scale of 1 to 9, with one being the most unpleasant and nine being the most pleasant. Exterior acceptance, section overview, taste, aroma, aftertaste, mouthfeel, elasticity, cohesiveness, juiciness, and overall acceptance were the attributes evaluated.

As shown in Figure 3, the expectedly overall acceptance score of the meatball samples treated with hot air convection ranged from 6 ± 0.5 to 8 ± 0.67 (*p* < 0.05), whereas the samples treated by steam convection received a score ranging from 5.5 ± 1.37 and 7.7 ± 0.67 (*p* < 0.05). Even though the color attribute of all the meatballs changed because of the thermal treatment, beef meat received the highest scores for exterior acceptance (ranging from 7.5 ± 0.53 to 8 ± 0.67) and overall consumer acceptance (between 7.6 ± 1.07 and 8.6 ± 0.52). Pork meatballs followed in terms of exterior acceptability; however, the turkey meatballs scored higher values (*p* < 0.05) for the overall acceptability by the panelists, after the beef ones. The assessment of color acceptance is critical because if the color is unacceptable, all other sensory attributes end up losing their significance in the eyes of consumers, resulting in a negative impact on their purchasing decisions [33].

## 3. Materials and Methods

### 3.1. Chemicals and Reagents

The following chemicals and reagents were used to perform all the determinations for the meatballs: 2,2-diphenyl-1-picryhydrazyl (DPPH), 6-hydroxy-2,5,7,8-tetramethylchromane-2-carboxylic acid (Trolox), potassium persulfate (K_2_O_2_S_8_), Folin–Ciocalteu reagent, gallic acid, sodium carbonate (Na_2_CO_3_) 20%, quercetin, sodium nitrite (NaNO_2_) 5%, aluminum chloride (AlCl_3_) 10%, sodium hydroxide (NaOH) 1M, chloroform, and methanol (HPLC grade), which were all purchased from Sigma-Aldrich Steinheim, Germany.

### 3.2. Samples Preparation

#### 3.2.1. Preparation of the Aqueous Extracts of Herbs

Aqueous extracts were obtained after the slightly modified method described by [34]. Briefly, 5 g of ground herbs were grinded (Gorenje grinder SMK150B, Ljubljana Republic of Slovenia) and mixed with 125 mL of bidistilled water. The samples were boiled on water bath for 30 min, then filtered, cooled, and refrigerated (4 °C).

#### 3.2.2. Preparation of Meatballs

Tenderloin from three distinct types of meat (pork, turkey, and beef) was purchased from a local market (Galați, Romania) and used to obtain meatballs. The meat was minced using a food processor (Philips HR 7766, Hungary) and mixed with aqueous extract of lemon balm or wild thyme (6%), sunflower oil (6%), salt (0.5%), and pepper (0.3%). Each meatball was manually modeled into spheres weighing 20 ± 1 g (3.5 ± 0.1 cm diameter) and subjected to heat treatment using an electric oven (Indesit FIMB-51K.A-IX-PL, Piekarnik, Poland) and a steam cooker (Zelmer 37Z010, Poznań, Poland). Steam convection was used for 15 min, while hot air convection was used for 36 min for pork meatballs and 30 min for turkey and beef meatballs, until the core temperature of minimum 72 °C was reached. Their spherical shape was selected based on the shape of the traditional meatballs consumed in European countries.

#### 3.2.3. Sample Codification of Herbal Aqueous Extract Enriched Meatballs

HP, HT, and HB were pork, turkey, and beef control meatballs, respectively, processed by hot air convection. SP, ST, and SB were pork, turkey, and beef control meatballs, respectively, thermally treated by steam convection.

ECPC and ECPA were pork meatballs enriched with wild thyme aqueous extract, treated by hot air and steam convection, respectively, while ERPC and ERPA were pork meatballs enriched with lemon balm aqueous extract, treated by hot air and steam convection, respectively.

ECCC and ECCA were turkey meatballs enriched with wild thyme aqueous extract, processed by hot air and steam convection, respectively. Meanwhile, ERCC and ERCA were turkey meatballs enriched with lemon balm aqueous extract, obtained by hot air and steam convection, respectively.

Finally, ECVC and ECVA were beef meatballs enriched with wild thyme aqueous extract, thermally treated by hot air and steam convection, respectively. As well, ERVC and ERVA were beef meatballs enriched with lemon balm aqueous extract, processed by hot air and steam convection, respectively.

#### 3.2.4. Preparation of Extraction

To determine the antioxidant activity and the total bioactive compounds, the method described by [35] was used, modified as follows. Briefly, 1 g of each sample was homogenized with 6 mL of distilled water and then centrifuged at 9000× *g* at 4 °C for 5 min, followed by the addition of 1.8 mL of chloroform.

### 3.3. Evaluation of Antioxidant Activity

The antioxidant activity was determined using DPPH-free radical scavenging assay, a modified method described by [36]. Therefore, 0.1 mL of extract was added in a test tube, followed by 3.9 mL of DPPH solution (0.1 M). For the control sample, the extract was replaced with 0.1 mL of methanol. The absorbance was determined spectrophotometrically at 515 nm with a UV–VIS spectrophotometer (Biochrom Libra S22, Cambridge, UK) after a 90 min incubation period at room temperature (20 °C), in the dark. The variation of the antioxidant capacity corresponding to the different samples was studied by determining µM Trolox/g for each sample to be analyzed.

### 3.4. Determination of Total Bioactive Compounds by Spectrophotometric Methods

#### 3.4.1. Total Phenolic Content (TPC)

The TPC was determined using the Folin–Ciocalteu method, taking as a reference the method described in [37]. Precisely, in a test tube, 0.20 mL of sample was mixed with 15.8 mL of distilled water, followed by 1 mL of Folin–Ciocalteu reagent. The sample was left to rest for 6 min in a dark place at room temperature. Then, 3 mL of 20% (*w*/*v*) solution prepared from sodium carbonate (Na_2_CO_3_) was added and the samples was allowed to rest for a 60 min incubation period at room temperature (21 ± 2 °C), in the dark. Thereafter, the absorbance was measured at a wavelength of 765 nm using an UV–VIS spectrophotometer (Biochrom Libra S22, Cambridge, UK). For the blank sample, in contrast to the prepared sample, methanol was used. The results are expressed as mg of gallic acid equivalents/g sample ± SD (mg GAE/g). All determinations were performed in triplicate.

#### 3.4.2. Total Flavonoid Content (TFC)

The TFC was determined using the method described by [38]. A volume of 0.25 mL of the sample was mixed with 1.25 mL of distilled water and, subsequently, with 0.075 mL of 5% NaNO_2_ solution. The mixture was allowed to react for 5 min, after which 0.15 mL of 10% AlCl_3_ solution was added and left to react again for 6 min. Finally, 0.5 mL of 1 M NaOH solution and 0.775 mL of distilled water were added. The absorbance of the resulting mixture was immediately read at a wavelength of 510 nm using a UV–VIS spectrophotometer (Biochrom Libra S22, Cambridge, UK). The TFC was determined using the standard quercetin curve and expressed as mg quercetin equivalents/mL sample ± SD (mg EQ/mL). All determinations were performed in triplicate.

### 3.5. Fourier-Transform Infrared Spectroscopy (FT-IR)

The infrared spectra were collected using a Nicolet iS50 FT-IR spectrometer (Thermo Scientific, OH, USA) equipped with a built-in ATR accessory, DTGS detector and KBr beamsplitter. Therefore, 32 scans were co-added over the range of 4000–400 cm^−1^, with a resolution of 4 cm^−1^. As a reference for the background spectrum, air was used before every sample. After each spectrum, the ATR plate was cleaned using ethanol solution. For validation that no residue from the previous sample remained, a background spectrum was collected each time and compared to the previous background spectrum. The FT-IR spectrometer rested in a room with air conditioned and controlled temperature (21 °C).

### 3.6. In Vitro Release of Phenolic Compounds

The protocol described by [39] was used to perform a simulated in vitro gastrointestinal digestion process for the meatballs, which imitates the gastric and intestinal phases. Briefly, gastric digestion was simulated using gastric juice (SGJ) with porcine pepsin (40 mg/mL in 0.1 M HCl, pH = 3.0). Simulated intestinal digestion was performed using intestinal juice (SIJ) containing pancreatin from porcine pancreas (2 mg/mL in 0.9 M baking soda, pH = 7). For each digestion phase, the samples were incubated for 2 h in an SI e300R orbital shaking incubator (Medline Scientific, Chalgrove, Oxon, UK), at 150 rpm and 37 °C. 0.2 mL of post-hydrolysis fraction was collected at every 30 min of digestion, centrifuged (14,000 rpm for 5 min), and analyzed for TPC content.

### 3.7. Color Measurements

The color parameters of the meatballs were performed using a MINOLTA Chroma Meter CR-410 (Konica Minolta, Osaka, Japan). For color analysis, each meatball was cut in half to have a uniform color. The color parameters determined were *L** (lightness/darkness), *a** (red/green), and *b** (yellow/blue). The total color difference (Δ*E*) between samples was calculated according to Equation (1):(1)$$\Delta E=\sqrt{{\left(L_{0}^{*}-L*\right)}^{2}+{\left(a_{0}^{*}-a*\right)}^{2}+{\left(b_{0}^{*}-b*\right)}^{2}}$$

Subscript 0 refers to the color of the non-heat treated sample. In addition, Chroma (*C**), color intensity, hue angle (*h**), and visual color appearance were calculated according to Equations (2) and (3):(2)$$C*=\sqrt{{a*}^{2}+{b*}^{2}}$$
(3)$$h*=tan^{-1}\left(\frac{b*}{a*}\right)$$

The whiteness index (*WI*) was calculated using *L**, *a**, and *b** values according to the following Equation (4):(4)$$WI=100-{\left[{\left(100-L*\right)}^{2}+{a*}^{2}+{b*}^{2}\right]}^{1/2}$$

In addition, the browning index (*BI*) and the yellowness index (*YI*) were calculated using Equations (5) and (6):(5)$$BI=100\times \left(\frac{X-0.31}{0.17}\right)$$
(6)where X=a*+1.75·L*5.645·L*+a*−3.012·b^ and YI=142.86·b*L*

All these parameters are dimensionless. The measurements were performed in triplicates.

### 3.8. Sensory Evaluation

Sensorial analysis was performed at room temperature (20 °C), under white light by a group of 10 untrained panelists belonging to Food Science and Engineering staff (two males and eight females). After each sample, the participants rinsed their mouths with plain water. The members of the panel group were non-smokers, aged between 24 and 54 years old. The sample preparation method and its benefits for the human body were explained before participation. Samples were served in a random order to each panelist on a white dish, coded with random numbers. The intensity of each attribute was determined using a hedonic nine-points scale, where 1 is the weakest/most unpleasant perception and 9 is the strongest/most pleasant perception. The tasters pointed the following sensorial attributes: exterior acceptance, section overview, taste, aroma, aftertaste, mouthfeel, elasticity, cohesiveness, juiciness, and overall appreciation.

### 3.9. Statistical Analysis

Three samples per treatment were analyzed and data were reported as mean ± standard deviation (SD). To determine the effect of treatments on the studied samples, a one-way analysis of variance (ANOVA) was performed. A Tukey test with a 95% confidence level was applied for post-hoc analysis; *p* < 0.05 was considered to be statistically significant. The statistical analysis was carried out using Minitab 19 statistical software (free trial).

## 4. Conclusions

The results of the present study indicate that herbal aqueous extracts have a statistically significant impact (*p* < 0.05) on the phytochemical content of the meatballs. Lemon balm enriched meatballs, processed by hot air convection, were characterized by higher content of polyphenols, whereas those obtained by steam convection showed a raised concentration of flavonoids. The best colorimetric results were shown among beef and pork meatballs, treated by hot air convection, making them the most appreciated by panelists. FT-IR analysis revealed the existence of multiple similar peaks within both lactation adjuvants (trans-anethole and estragole) and the meatballs supplemented with aqueous extracts. The findings showed that herbal aqueous extracts reveal interesting possibilities for the development of functional foods with special destination, both in terms of the lactogenic activity determined by FT-IR analysis and other beneficial properties that contribute to the wellbeing of the human body.

## Figures and Tables

**Figure 1 molecules-27-03920-f001:**
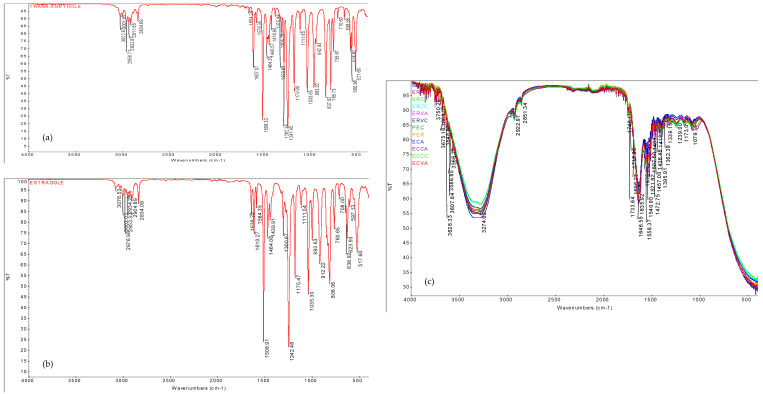
Fourier-transform infrared spectroscopy spectra of (**a**) standard trans-anethole, (**b**) standard estragole, (**c**) meatballs with the addition of aqueous extract of lemon balm/wild thymes; ECPC and ECPA are pork meatballs enriched with wild thyme aqueous extract, processed by hot air and steam convection; ERPC and ERPA are pork meatballs enriched with lemon balm aqueous extract, processed by hot air and steam convection; ERCC and ERCA are turkey meatballs enriched with lemon balm aqueous extract, processed by hot air and steam convection; ECVC and ECVA are beef meatballs enriched with wild thyme aqueous extract, processed by hot air and steam convection; ERVC and ERVA are beef meatballs enriched with lemon balm aqueous extract, processed by hot air and steam convection.

**Figure 2 molecules-27-03920-f002:**
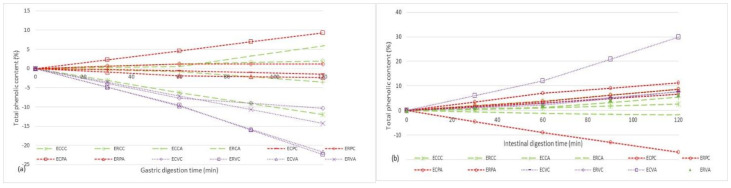
The in vitro digestion of remanent polyphenolic content in meatball samples in simulated gastric juice (**a**) and simulated intestinal juice (**b**); ECPC and ECPA are pork meatballs enriched with wild thyme aqueous extract, processed by hot air and steam convection; ERPC and ERPA are pork meatballs enriched with lemon balm aqueous extract, processed by hot air and steam convection; ERCC and ERCA are turkey meatballs enriched with lemon balm aqueous extract, processed by hot air and steam convection; ECVC and ECVA are beef meatballs enriched with wild thyme aqueous extract, processed by hot air and steam convection; ERVC and ERVA are beef meatballs enriched with lemon balm aqueous extract, processed by hot air and steam convection.

**Figure 3 molecules-27-03920-f003:**
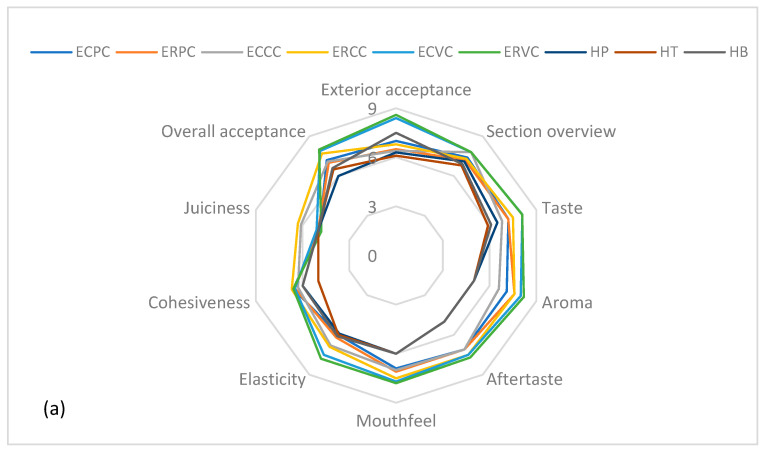
Comparative diagram of the sensory attributes specific to meatballs, obtained by (**a**)—hot air convection and (**b**)—steam convection; ECPC and ECPA are pork meatballs enriched with wild thyme aqueous extract; ERPC and ERPA are pork meatballs enriched with lemon balm aqueous extract; ERCC and ERCA are turkey meatballs enriched with lemon balm aqueous extract; ECVC and ECVA are beef meatballs enriched with wild thyme aqueous extract; ERVC and ERVA are beef meatballs enriched with lemon balm aqueous extract; HP, HT and HB are pork, turkey, and beef control meatballs, processed by hot air convection. SP, ST, and SB are pork, turkey, and beef control meatballs thermally treated by steam convection.

**Table 1 molecules-27-03920-t001:** Phytochemical profile of herbal aqueous extract enriched meatballs.

	*Hot Air Convection*								
	HP	ERPC	ECPC	HT	ECCC	ERCC	HB	ECVC	ERVC
**Antioxidant Activity, µM Trolox/g dw**	5.90 ± 0.01 ^B^	7.50 ± 0.22 ^A^	3.36 ± 0.06 ^D,E^	3.20 ± 0.02 ^D,E^	3.53 ± 0.27 ^C,D^	3.97 ± 0.24 ^C,D^	2.32 ± 0.00 ^E^	3.12 ± 0.64 ^D,E^	4.56 ± 0.76 ^C^
**TPC, mg GAE/g dw**	1.80 ± 0.00 ^E^	2.21 ± 0.03 ^B^	1.54 ± 0.03 ^F^	1.71 ± 0.00 ^E,F^	1.79 ± 0.03 ^E^	1.82 ± 0.02 ^D,E^	2.16 ± 0.00 ^B,C^	2.00 ± 0.03 ^C,D^	2.69 ± 0.02 ^A^
**TFC, mg EQ/g dw**	1.06 ± 0.00 ^B^	1.51 ± 0.19 ^A^	1.59 ± 0.03 ^A^	1.66 ± 0.00 ^A^	1.76 ± 0.10 ^A^	1.80 ± 0.20 ^A^	1.77 ± 0.00 ^A^	1.73 ± 0.05 ^A^	1.73 ± 0.05 ^A^
	** *Steam Convection* **								
	**SP**	**ECPA**	**ERPA**	**ST**	**ECCA**	**ERCA**	**SB**	**ECVA**	**ERVA**
**Antioxidant Activity, µM Trolox/g dw**	0.11 ± 0.03 ^F^	1.00 ± 0.04 ^E,F^	1.40 ± 0.09 ^E^	4.5 ± 0.00 ^C^	3.99 ± 0.19 ^C,D^	3.05 ± 0.64 ^D^	4.86 ± 0.01 ^B,C^	5.99 ± 0.69 ^A^	5.71 ± 0.45 ^A,B^
**TPC, mg GAE/g dw**	1.81 ± 0.00 ^A^	1.57 ± 0.04 ^B,C^	1.56 ± 0.01 ^B,C^	1.68 ± 0.00 ^A,B^	1.72 ± 0.05 ^A,B^	1.75 ± 0.04 ^A^	1.47 ± 0.00 ^C^	1.46 ± 0.02 ^C^	1.38 ± 0.04 ^C^
**TFC, mg EQ/g dw**	1.93 ± 0.00 ^C,D^	1.88 ± 0.07 ^C,D^	2.49 ± 0.14 ^B^	1.7 ± 0.00 ^D^	1.89 ± 0.14 ^C,D^	1.72 ± 0.02 ^D^	1.58 ± 0.00 ^D^	2.22 ± 0.20 ^B,C^	3.03 ± 0.24 ^A^

The averages on the same line that do not share a super script (A–F) are statistically significantly different (*p* < 0.05); Total Phenolic Content (TPC); Total Flavonoid Content (TFC); ECPC and ECPA are pork meatballs enriched with wild thyme aqueous extract; ERPC and ERPA are pork meatballs enriched with lemon balm aqueous extract; ERCC and ERCA are turkey meatballs enriched with lemon balm aqueous extract; ECVC and ECVA are beef meatballs enriched with wild thyme aqueous extract; ERVC and ERVA are beef meatballs enriched with lemon balm aqueous extract; HP and SP are control pork meatballs obtained by hot air and steam convection; HT and ST are control turkey meatballs obtained by hot air and steam convection; HB and SB are control beef meatballs obtained by hot air and steam convection.

**Table 2 molecules-27-03920-t002:** Effects of heat treatment on the color parameters of aqueous extract enriched meatballs.

Samples Color Parameters	Hot Air Convection						Steam Convection					
	ECPC	ERPC	ECCC	ERCC	ECVC	ERVC	ECPA	ERPA	ECCA	ERCA	ECVA	ERVA
** *L** **	50.47 ± 2.26 ^B^	51.37 ± 3.76 ^B^	57.58 ± 0.33 ^A^	57.75 ± 0.29 ^A^	42.08 ± 0.64 ^C^	40.93 ± 0.59 ^C^	48.55 ± 0.62 ^C^	49.34 ± 0.21 ^C^	57.93 ± 0.18 ^A^	56.47 ± 0.55 ^B^	41.69 ± 1.07 ^D^	41.37 ± 0.60 ^D^
** *a** **	4.65 ± 0.12 ^B^	4.83 ± 0.48 ^B^	2.68 ± 0.02 ^C^	2.30 ± 0.01 ^C^	6.99 ± 0.30 ^A^	6.97 ± 0.09 ^A^	4.83 ± 0.04 ^C^	4.78 ± 0.07 ^C^	2.10 ± 0.04 ^D^	2.11 ± 0.01 ^D^	6.39 ± 0.16 ^A^	6.12 ± 0.04 ^B^
** *b** **	7.33 ± 0.11 ^A^	7.39 ± 0.03 ^A^	6.55 ± 0.09 ^B,C^	6.44 ± 0.21 ^C^	6.77 ± 0.14 ^B^	7.15 ± 0.14 ^A^	6.37 ± 0.11 ^B^	6.34 ± 0.28 ^B^	7.02 ± 0.02 ^A^	7.28 ± 0.17 ^A^	7.02 ± 0.11 ^A^	7.25 ± 0.01 ^A^
**Δ*E***	19.38 ± 2.09 ^A^	17.18 ± 3.32 ^A^	18.29 ± 0.29 ^A^	18.26 ± 0.28 ^A^	16.39 ± 0.42 ^A^	12.20 ± 0.49 ^B^	17.48 ± 0.60 ^B^	15.29 ± 0.23 ^D^	18.83 ± 0.18 ^A^	17.20 ± 0.47 ^B,C^	16.44 ± 0.81 ^C^	13.09 ± 0.42 ^E^
** *C** **	8.68 ± 0.16 ^B^	8.83 ± 0.24 ^B^	7.07 ± 0.08 ^C^	6.83 ± 0.20 ^C^	9.73 ± 0.31 ^A^	9.98 ± 0.17 ^A^	7.99 ± 0.12 ^B^	7.94 ± 0.27 ^B^	7.32 ± 0.01 ^C^	7.58 ± 0.16 ^C^	9.49 ± 0.19 ^A^	9.49 ± 0.02 ^A^
** *h** **	−0.01 ± 0.02 ^B^	0.03 ± 0.03 ^B^	−1.20 ± 0.13 ^C^	−2.93 ± 0.76 ^D^	0.69 ± 0.03 ^A^	0.61 ± 0.01 ^A,B^	0.26 ± 0.01 ^C^	0.25 ± 0.04 ^C^	5.35 ± 2.13 ^A^	3.13 ± 0.73 ^B^	0.51 ± 0.01 ^C^	0.41 ± 0.01 ^C^
** *WI* **	49.71 ± 2.19 ^B^	50.57 ± 3.66 ^B^	56.99 ± 0.33 ^A^	57.20 ± 0.32 ^A^	41.26 ± 0.58 ^C^	40.09 ± 0.61 ^C^	47.93 ± 0.63 ^C^	48.72 ± 0.24 ^C^	57.30 ± 0.18 ^A^	55.82 ± 0.57 ^B^	40.92 ± 1.03 ^D^	40.60 ± 0.59 ^D^
** *BI* **	22.12 ± 0.62 ^C^	22.13 ± 1.11 ^C^	15.20 ± 0.24 ^D^	14.47 ± 0.48 ^D^	29.25 ± 0.43 ^B^	21.22 ± 1.06 ^A^	21.03 ± 0.61 ^B^	20.53 ± 0.85 ^B^	15.28 ± 0.06 ^C^	16.24 ± 0.52 ^C^	29.22 ± 0.22 ^A^	29.70 ± 0.43 ^A^
** *YI* **	20.76 ± 0.61 ^C^	20.61 ± 1.59 ^C^	16.24 ± 0.32 ^D^	15.92 ± 0.59 ^D^	22.99 ± 0.13 ^B^	24.97 ± 0.85 ^A^	18.75 ± 0.57 ^B^	18.36 ± 0.90 ^B^	17.30 ± 0.00 ^C^	18.24 ± 0.61 ^B^	24.04 ± 0.26 ^A^	25.04 ± 0.41 ^A^

The averages on the same line with different superscripts (A, B, C and D) are statistically significantly different (*p* < 0.05). *L**—clarity/brightness; *a**—red / green color component; *b**—blue / yellow color component; Δ*E*—total color difference; *C**—chrome; *h**—hue angle; *BI*—browning index; *WI*—whitening index; *YI*—yellowness index; ECPC and ECPA are pork meatballs enriched with wild thyme aqueous extract; ERPC and ERPA are pork meatballs enriched with lemon balm aqueous extract; ERCC and ERCA are turkey meatballs enriched with lemon balm aqueous extract; ECVC and ECVA are beef meatballs enriched with wild thyme aqueous extract; ERVC and ERVA are beef meatballs enriched with lemon balm aqueous extract.

**Table 3 molecules-27-03920-t003:** Effects of heat treatment on the color parameters of one week refrigerated meatballs.

Samples Color Parameters	Hot Air Convection						Steam Convection					
	ECPC	ERPC	ECCC	ERCC	ECVC	ERVC	ECPA	ERPA	ECCA	ERCA	ECVA	ERVA
** *L** **	49.68 ± 0.76 ^B^	50.43 ± 0.42 ^B^	57.30 ± 0.20 ^A^	56.65 ± 0.39 ^A^	41.75 ± 0.57 ^A^	41.19 ± 0.56 ^A^	49.69 ± 0.45 ^B^	49.32 ± 0.12 ^B^	57.36 ± 1.68 ^A^	55.97 ± 0.62 ^A^	41.75 ± 0.85 ^C^	40.51 ± 0.26 ^C^
** *a** **	3.23 ± 0.02 ^B^	2.70 ± 0.00 ^C^	1.93 ± 0.07 ^D^	1.58 ± 0.06 ^E^	3.59 ± 0.08 ^A^	3.56 ± 0.01 ^A^	2.80 ± 0.01 ^D^	2.99 ± 0.13 ^C^	1.16 ± 0.02 ^E^	1.32 ± 0.06 ^E^	3.50 ± 0.09 ^B^	3.67 ± 0.12 ^A^
** *b** **	7.48 ± 0.14 ^A^	7.35 ± 0.06 ^A,B^	7.16 ± 0.19 ^B,C^	6.91 ± 0.18 ^C^	7.38 ± 0.08 ^A,B^	7.53 ± 0.01 ^A^	7.96 ± 0.18 ^A^	7.76 ± 0.05 ^A^	7.69 ± 0.25 ^A^	7.84 ± 0.06 ^A^	6.89 ± 0.57 ^B^	7.45 ± 0.01 ^A^
**Δ*E***	19.18 ± 0.69 ^A^	17.18 ± 0.37 ^C^	18.35 ± 0.20 ^A,B^	17.68 ± 0.30 ^B,C^	18.21 ± 0.49 ^B^	14.83 ± 0.35 ^D^	19.41 ± 0.40 ^A^	16.07 ± 0.04 ^C^	18.78 ± 1.44 ^A^	17.22 ± 0.45 ^B,C^	18.16 ± 0.44 ^A,B^	14.31 ± 0.25 ^D^
** *C** **	8.15 ± 0.14 ^A^	7.83 ± 0.05 ^B^	7.41 ± 0.20 ^C^	7.09 ± 0.17 ^D^	8.20 ± 0.03 ^A^	8.32 ± 0.01 ^A^	8.43 ± 0.17 ^A^	8.00 ± 0.10 ^A,B,C^	7.77 ± 0.24 ^C^	7.95 ± 0.07 ^B,C^	7.73 ± 0.47 ^C^	8.30 ± 0.07 ^A,B^
** *h** **	−0.93 ± 0.05 ^D^	−2.25 ± 0.13 ^E^	1.58 ± 0.13 ^A^	0.36 ± 0.31 ^B^	−0.53 ± 0.09 ^C^	−0.60 ± 0.01 ^C^	−3.27 ± 0.56 ^B^	−1.33 ± 0.26 ^B^	4.38 ± 4.17 ^A^	−4.45 ± 3.64 ^B^	−0.44 ± 0.26 ^B^	−0.50 ± 0.08 ^B^
** *WI* **	49.02 ± 0.72 ^B^	49.82 ± 0.43 ^B^	56.66 ± 0.25 ^A^	56.07 ± 0.41 ^A^	41.18 ± 0.56 ^C^	40.60 ± 0.55 ^C^	48.99 ± 0.47 ^B^	48.69 ± 0.10 ^B^	56.65 ± 1.61 ^A^	55.26 ± 0.60 ^A^	41.24 ± 0.90 ^C^	39.93 ± 0.27 ^C^
** *BI* **	20.75 ± 0.02 ^C^	19.36 ± 0.30 ^D^	15.52 ± 0.38 ^E^	14.77 ± 0.40 ^F^	25.38 ± 0.30 ^B^	26.15 ± 0.34 ^A^	21.24 ± 0.65 ^C^	20.43 ± 0.26 ^C^	15.55 ± 0.03 ^D^	16.49 ± 0.00 ^D^	23.84 ± 1.99 ^B^	26.59 ± 0.45 ^A^
** *YI* **	21.51 ± 0.08 ^C^	20.82 ± 0.34 ^D^	17.84 ± 0.39 ^E^	17.43 ± 0.58 ^E^	25.24 ± 0.08 ^B^	26.10 ± 0.33 ^A^	22.87 ± 0.72 ^B,C^	21.51 ± 0.09 ^C,D^	19.14 ± 0.06 ^E^	20.01 ± 0.08 ^D,E^	23.58 ± 2.44 ^B^	26.28 ± 0.22 ^A^

The averages on the same line with different superscripts (A, B, C, D, E and F) are statistically significantly different (*p* < 0.05). *L**—clarity/brightness; *a**—red/green color component; *b**—blue/yellow color component; Δ*E*—total color difference; *C**—chrome; *h**—hue angle; *BI*—browning index; *WI*—whitening index; *YI*—yellowness index; ECPC and ECPA are pork meatballs enriched with wild thyme aqueous extract; ERPC and ERPA are pork meatballs enriched with lemon balm aqueous extract; ERCC and ERCA are turkey meatballs enriched with lemon balm aqueous extract; ECVC and ECVA are beef meatballs enriched with wild thyme aqueous extract; ERVC and ERVA are beef meatballs enriched with lemon balm aqueous extract.

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
