# Peer review of "Different Types of Meatballs Enriched with Wild Thyme/Lemon Balm Aqueous Extract—Complex Characterization"

_molecules, 2022, doi:10.3390/molecules27123920_

Round 1

Reviewer 1 Report

Review of MS: molecules-1755541 entitled “Different types of meatballs enriched with wild thyme/lemon balm aqueous extract – complex characterization”

General Comments

This paper has seriuos flaws in the experimental part and in conceptualization. Authors propose to add thyme and lemon balm aqueous extracts to meatballs with the aim to ameliorate lactation in women. In addition they made a “complex characterization” whose concept is not clear.

It is not clear why acqueous extract were added to meatballs; can these extract be assumed as they are? Authors did not make any kind of optimization for the product they obtained; infact “Costumers overall acceptance had a score above 5.5 for all samples, on a 1 to 9 scale”. In my opinion this score is not good for a commercial product.

“Two thermal treatments, such as hot air convection (180°C) and steam convection (94°C), were applied for meatballs processing”. Why Authors choose these conditions for meatballs processing? Were these conditions optimized?

“Further analysis and human trials should be considered regarding the use of lactogenic herbs, given their health benefits and availability” Authors conclude in the Abstract that further analyses are needed to demonstrate the effect of addition of extracts to meatballs!

“FT-IR analysis confirmed the presence of trans-anethole and estragole at 1507-1508 cm-1 and 1635-1638 cm-1, respectively”. FT-IR analysis is an unusal method to asses the presence of compounds in complex matrices. In this case Authors conclude with presence of trans-anethole and estragole only by using FT-IR. In my opinion Authors must use cromatography techniques to identify compounds. The use of HPLC is required for polyphenol analysis too.

“Lemon balm enriched meatballs, processed by hot air convection, were characterized by higher content of polyphenols, whereas those obtained by steam convection showed raised concentration of flavonoids”. Colorimetric tests are not sufficient to assess different contents of polyphenols and so instrumental analyses are required.

Line 212-290. Authors discuss of changing of color of meatballs and show two tables. In my opinion this parameter should be taken in consideration after the optimization of all parameters. More important parameter is shelf life of product that is mainly based on microbiolgical analyses.

For these reasons I suggest to not publish the paper in this form.

Author Response

The authors would like to thank for the suggestions and the spent time to review the manuscript. We are hopping that the answers are suitable to reconsider your opinion and to clarify some scientific aspects.

Reviewer 2 Report

First of all, explain what does mean „complex“ characterization?

Regarding „healthy food products, “ authors probably meant functional food that provides beneficial health effects in addition to nutrient value. There are no healthy and non-healthy foods, but only healthy and unhealthy eating habits from a nutritional science point of view. So, rephrase the first sentence in the Abstract. In addition, the authors stated, „ functionality plant extracts was proved by the values of the total content of polyphenols and flavonoids. “ The contents of bioactive compounds themselves do not prove functionalities.

Regarding the aim of this study, there are no clear do herbal extracts added to meatballs to prevent lipid peroxidation or provide lactogenic potential?

The last sentence in the Introduction is more appropriate for the Conclusion section.

The additional aim was to investigate the effects of different processing methods in enriched meatballs?

Why, in addition to the DPPH test, did the authors not evaluate any lipid peroxidation parameters?

In the Results and Discussion (lines 96 to 106 and lines 236-240), there is no need to repeat the results from the tables in the text. The text should point out key findings or trends, not repeat previously presented data. Please correct that.

Why did the discussion not contain some limitations regarding herbal extracts in food products, especially for breastfeeding women?

Define the author's contribution roles (lines 497-500)

Author Response

The authors would like to thank for the suggestions and the spent time to review the manuscript. We are glad that you have contribute to the improvement of our article.

Reviewer 3 Report

The article “Different types of meatballs enriched with wild thyme/lemon balm aqueous extract – complex characterization” is well written and needs to be published after minor revision. The comments need to be filled before acceptance are:

1). The abstract defines the meat ball with wild thyme/lemon balm is for lactation problems among breastfeeding women. kindly define the how it is good for lactation problems woman.

2). Modify the objective part in the article

3). Modify FTIR region to justify work

4). Conclusion needs minor revision

Author Response

(The authors gave the same response as above.)

Round 2

Reviewer 1 Report

Authors have improved the manuscript accepting all comments moved by reviewers. For this reason I suggest to accept manuscript for publication in this form.